# UNLOCKING THE POWER OF MIXTURE-OF-EXPERTS FOR TASK-AWARE TIME SERIES ANALYTICS

## ABSTRACT

Time Series Analysis is widely used in various real-world applications such as weather forecasting, financial fraud detection, imputation for missing data in IoT systems, and classification for action recognition. Mixture-of-Experts (MoE), as a powerful architecture, though demonstrating effectiveness in NLP, still falls short in adapting to versatile tasks in time series analytics due to its task-agnostic router and the lack of capability in modeling channel correlations. In this study, we propose a novel, general MoE-based time series framework called PatchMoE to support the intricate "knowledge" utilization for distinct tasks, thus task-aware. Based on the observation that hierarchical representations often vary across tasks, e.g., forecasting vs. classification, we propose a Recurrent Noisy Gating to utilize the hierarchical information in routing, thus obtaining task-sepcific capability. And the routing strategy is operated on time series tokens in both temporal and channel dimensions, and encouraged by a meticulously designed Temporal & Channel Load Balancing Loss to model the intricate temporal and channel correlations. Comprehensive experiments on five downstream tasks demonstrate the state-of-the-art performance of PatchMoE.

**Resources:** https://anonymous.4open.science/r/PatchMoE-BD38.

## 1 INTRODUCTION

Time Series Analysis is widely used in real-world applications, with key tasks such as forecasting (Cirstea et al., 2022; Qiu et al., 2025b), anomaly detection (Wu et al., 2025b; Wang et al., 2023a), imputation (Tashiro et al., 2021) and classification (Chen et al., 2025), among others (Wu et al., 2024b;a), gaining attention. In recent years, many deep-learning networks are proposed for these specific tasks, and achieve great progress. Most of them feature distinct meticulously-designed representation learning backbones, aiming at capturing task-specific inductive bias within data, and actually outperform those general algorithms (Wu et al., 2023; Nie et al., 2023; Liu et al., 2024c). Therefore, *there still lacks a general and powerful enough backbone to explicitly and effectively capture the task-specific characteristics in different time series tasks,* like ResNet in CV and GPT in NLP. Mixture-of-Experts (MoE) (Shazeer et al., 2017; Aljundi et al., 2017), as a powerful framework, is widely applied in CV and NLP, and proven effective and efficient by activating different experts to solve problems from different distributions, possessing the potential of exceling at all tasks. However, there still exists some challenges in adapting MoE to time series analysis.

In Time Series Analytics, some studies (Wu et al., 2023; Liu et al., 2024c; Luo & Wang, 2024; Nie et al., 2023) reveal the phenomenon that CKA (centered kernel alignment (Cortes et al., 2012)) similarities of the representations from the first and last layers often show distinguishable differences in different tasks of time series analytics. As shown in Figure 1, stronger models often show higher CKA similarities in forecasting and anomaly detection, and lower CKA similarities in imputation and classification. *This indicates key task-specific characteristics exist in representations of different levels, and well-performed models (like PatchTST, iTransformer) can **implicitly** adapt the hierarchical representations in different layers to extract the task-specific characteristics.* However, since the "predict next token" paradigm has unified all language tasks of NLP, advanced MoE architectures (Liu et al., 2024a; Ma et al., 2018) may not consider such task-specific hierarchical representational differences during routing, *thus limiting the ability of explicitly utilizing task-specific characteristics across layers for time series analytics.*

Moreover, the Channel-Independent Transformer (Nie et al., 2023), as a basic structure insensitive to the number of channels and input lengths, has been used in many applications (Liu et al., 2024d; Woo et al., 2024; Liu et al., 2024b; 2025), appropriate to be integrated with MoEs.

While, due to the Channel-Independent (CI) Strategy, it lacks the ability to model the intricate channel and temporal correlations. Due to the univariate property in NLP, recent advanced MoE architectures also *cannot perform channel-wise routing for them and still follows CI*, thus hindering capturing the channel correlations. Therefore, this calls for a mechanism to capture these correlations to adapt MoE in transformers for time series analytics.

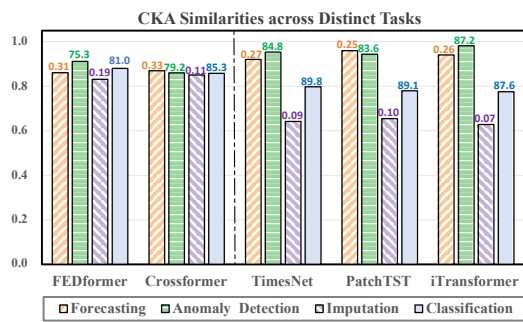

To handle the aforementioned limitations, integrating the MoE architecture with transformers and making it possess the task-specific capability while capturing the channel correlation provides an elegant solution for time series analytics. Intuitively, we propose a framework called **PatchMoE**. As its core component, the Recurrent Noisy Gating (RNG-Router) can dynamically perceive the representational differences across layers to model the hierarchical conditional probability distributions in the routing strategy, thus effectively routing experts to extract knowledge for distinct tasks. Moreover, time series tokens from different channels and timestamps are simultaneously routed to cap-

Figure 1: Representation analytics in Forecasting (Weather input-96-predict-336; MSE), Anomaly Detection (SMD; F1-Score), Imputation (Electricity Mask 37.5%; MSE), and Classification (PEMS-SF; Accuracy). For each model, we calculate the CKA similarity (refer to the vertical axis corresponding to the columns) between representations from the first and the last layers, and mark the performance of each task at the top of columns. Stronger models show more *distinguishable* CKA simiarlities across different tasks.

ture the intricate temporal and channel correlations. We also design the Temporal & Channel Load Balancing Loss to guide the MoE to model the sparse correlations, which is a better strategy (Qiu et al., 2025b; Wu et al., 2025b) between CI and CD. Inspired by recent works from multiple domains (Liu et al., 2024a; Ma et al., 2018; Fedus et al., 2022), we realize that applying the MoE architecture in the basic architecture to replace the FeedForward layer in CI-based Transformers may effectively utilize the knowledge and tackle tasks of time series analysis. Specifically, we use shared experts to extract common temporal patterns and routed experts to extract the differences among temporal and channel representations, so as to better model complex and distinct downstream tasks. Our contributions lie in:

- We propose a cross-task framework called PatchMoE for time series analysis. It can effectively utilize the hierarchical representational information for knowledge extraction, and enhance the CI-based Transformers in modeling intricate temporal and channel correlations.

- We devise a Recurrent Noisy Gating to effectively route experts based on the hierarchical representations for different tasks, which can enhance the performance of distinct downstream tasks.

- We propose the Temporal & Channel Load Balancing Loss to encourage the modeling of sparse correlations, which leads to a better temporal and channel strategy.

- As a general framework supporting multiple tasks, PatchMoE demonstrates consistent state-of-the-art performance on forecasting, anomaly detection, imputation and classification.

## 2 RELATED WORKS

### 2.1 TIME SERIES ANALYTICS

In recent years, time series analytics gain sustained attention. In forecasting, most works such as CNNs (Wu et al., 2023; Luo & Wang, 2024; Wang et al., 2022), MLPs (Lin et al., 2024b;a; Xu et al., 2024; Li et al., 2023), and Transformers (Cirstea et al., 2022; Nie et al., 2023; Dai et al., 2024; Zhang & Yan, 2022) manage to capture periodicity and trends within data and achieve good performance. In anomaly detection, reconstruction-based (Wu et al., 2025b; Nam et al., 2024) methods show

strong capabilities in detecting heterogeneous anomalies, and applying time-frequency analysis can effectively enhance the detection of subsequence anomalies. In imputation, capturing the underlying structures and complex temporal dynamics of time series data is important. By learning the true data distribution from observed values, deep learning imputation methods (Gao et al., 2025; Wang et al.; Du et al., 2023) can generate more reliable missing data. For classification, constrative learning methods (Wang et al., 2023b; Eldele et al., 2023; Chen et al., 2025) are widely used to construct the positive and negative pairs based on prior knowledge, which enhances the representation capability of encoders to identify different types of sequences.

## 2.2 MIXTURE-OF-EXPERTS

The mixture of experts (MoE) has been comprehensively explored and advanced, as demonstrated by subsequent studies (Shazeer et al., 2017; Aljundi et al., 2017; Zhou et al., 2022b). As the most important component, the routing mechanism of MoE gains wide attention. Noisy Gating (Shazeer et al., 2017) and Multi-Gating (Ma et al., 2018) are widely used to stablize the training and have many variations, but they do not consider task-specific information during routing. The load balancing constraint (Liu et al., 2024a; Shazeer et al., 2017) is also important, lots of task-specific optimization objectives are designed to mitigate the imbalance phenomenon in routing strategy, but they lack the generalization in time series analysis when facing multivariate modeling. For the basic architecture, most recent methods (Liu et al., 2024a; Ma et al., 2018; Riquelme et al., 2021) give priority to sparse MoE rather than dense MoE. As a modular layer, MoE demonstrates its flexibility and effectiveness in multiple real-world applications (Riquelme et al., 2021; Liu et al., 2024a; Ma et al., 2018), and the most common use is to replace the FeedForward layer in Transformer, which is generally believed to store and utilize the "knowledge". Famous works such as Switch Transformer (Fedus et al., 2022), Llama (Touvron et al., 2023), DeepSeek (Liu et al., 2024a), and MMoE (Ma et al., 2018) all follow this paradigm. In time series analytics, though some works (Liu et al., 2024b; Shi et al., 2024; Chen et al., 2024) apply the MoE layers in their models, no specific MoEs are devised for time series analysis to fully utilize the task-wise inductive bias within data. In this study, PatchMoE adopts a novel MoE structure tailored for task-specific representation learning, and can model intricate temporal and channel correlations.

## 3 METHODOLOGY

### 3.1 STRUCTURE OVERVIEW

As demonstrated in Figure 2, our proposed **PatchMoE** introduces a novel Mixture-of-Experts (MoE) framework. We reinforce the feedward layers with PatchMoE to effectively extract and utilize the "knowledge" from high-dimensional hidden representations. A multivariate time series is first processed through Normalization & Tokenization–see Section 3.2 to form the time series tokens. The tokens are then fed into Transformer layers to further extract the hidden semantics. In the MoE layer, the RNG-Router–see Section 3.3 models the conditional distribution of current routing strategy with a Recurrent Noisy Gating, which can integrate the representations from pre-layers, thus considering the main differences of various downstream tasks. Subsequently, the multivariate time series tokens are routed simultaneously to model the temporal and channel correlations. Specifically, we design the Temporal & Channel Load Balancing Loss–see Section 3.4 to encourage the RNG-Router adaptively route tokens with similar temporal or channel patterns into the same group of experts. The loss function encourages the green cases and mitigates the red cases in Figure 2 right. Considering the basic architecture, we also adpot the novel expert framework inspired by DeepSeek (Liu et al., 2024a), with Shared Experts and Routed Experts–see Section 3.5. The Shared Experts are designed to capture the general patterns in time series tokens, and the routed experts are assigned by the RNG-Router to flexibly construct the temporal and channel correlations. Finally, after the Transformer layers learn the representations, the task heads make outputs for different tasks, i.e., forecasting, anomaly detection, imputation, and classification.

### 3.2 NORMALIZATION & TOKENIZATION

The statistical property of time series varies over the time and causes distributional shift which hinders the performance of downstream tasks. For multivariate time series $X \in \mathbb{R}^{N \times T}$ with $N$ variates

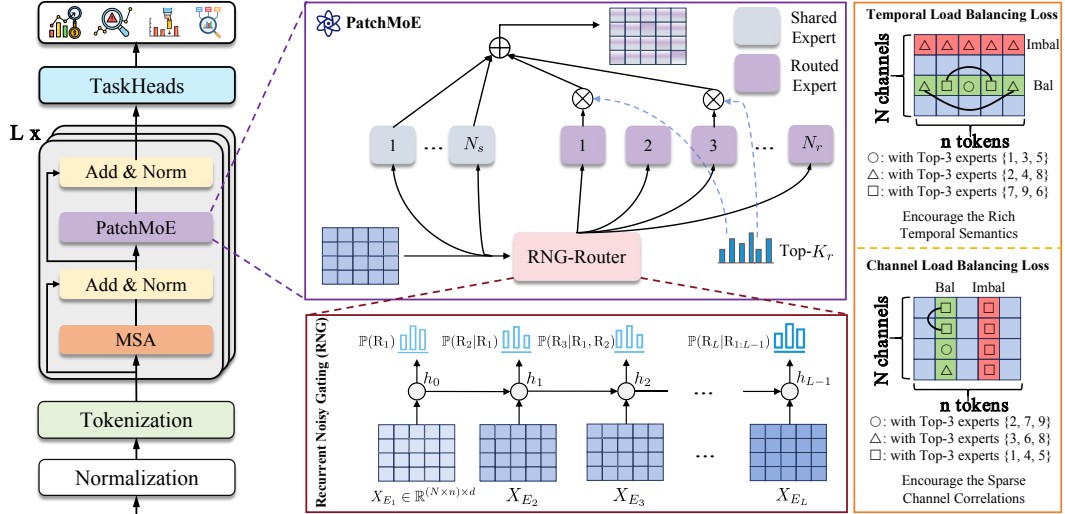

Figure 2: The overview of PatchMoE. The time series is first normalized and tokenized to make time series "tokens". In the $L$-stacked Transformer layers, the time series tokens are then processed through Multi-head Self-Attention (MSA) mechanism to obtain the representations. In the $l$-th layer, the RNG-Router takes the $X_{E_l} \in \mathbb{R}^{(N \times n) \times d}$ and the hidden state $h_{l-1} \in \mathbb{R}^{(N \times n) \times d}$ as inputs, utilizes the task-specific characteristics inside them to effectively route the experts. The Temporal & Channel Load Balancing Loss is designed to encourage the modeling of sparse temporal and channel correlations, which can enhance the temporal semantics and construct better Channel Strategies between CI and CD. See red and green tokens, encouraged by the Temporal & Channel Load Balancing Loss, green ones indicates that tokens are routed to different group of experts for balance.

and $T$ timestamps, PatchMoE adopts the Revin (Kim et al., 2021; Liu et al., 2022) technique for normalization to remove the varying statistical properties from the model's internal representations.

For the normalized time series $X_{norm} \in \mathbb{R}^{N \times T}$, we then utilize the Patching & Embedding technique (Cirstea et al., 2022; Nie et al., 2023; Wu et al., 2025b;a) for tokenization. The normalized time series is first divided into patches, and then projected into high-dimensional tokens:

$$X_P = \text{Patching}(X_{norm}) \in \mathbb{R}^{N \times n \times p}, \tag{1}$$

$$X_{token} = \text{Linear}(X_P) \in \mathbb{R}^{N \times n \times d}, \tag{2}$$

where $X_{token} \in \mathbb{R}^{N \times n \times d}$ are the embedded time series tokens. In the Multi-head Self-Attention (MSA) of Transformer layers, the tokens are further processed to extract the inherent temporal semantics:

$$X_E = \text{LayerNorm}(X_{token} + \text{MSA}(X_{token})), \tag{3}$$

where $X_E \in \mathbb{R}^{N \times n \times d}$ is the output of MSA. Note that in the MSA, the $X_{token}$ is processed in a Channel-Independent manner, where the channel correlations are not considered.

## 3.3 RNG-ROUTER

In the MoE layer, the processed tokens $X_E \in \mathbb{R}^{N \times n \times d}$ are first fed into the RNG-Router to decide which group of experts are activated for each token. The RNG-Router is based on the Recurrent Noisy Gating (RNG) mechanism, which models the conditional normal distribution of current routing strategy. This design utilizes the hierarchical information from Transformer layers to enhance the task-specific capabilities of PatchMoE, and stablizes the training process through a probability sampling paradigm (Shazeer et al., 2017). It is noted that the hierarchical information means the outputs of MSA layers in $L$ stacked Transformer layers and are denoted as $\{X_{E_1}, X_{E_2}, \cdots, X_{E_L}\}$. As aforementioned, these representations show distinct characteristics in different downstream tasks so that considering them into routing strategy to better extract the knowledge is rational.

Intuitively, we make the Recurrent Noisy Gating shared by all $L$ MoE layers of the $L$-stacked Transformer layers. In the $l$-th MoE layer, the Recurrent Noisy Gating takes the $l$-th MSA's output $X_{E_l} \in \mathbb{R}^{(N \times n) \times d}$ and hidden state $h_{l-1} \in \mathbb{R}^{(N \times n) \times d}$ from the previous layer as inputs, outputs $O_l \in \mathbb{R}^{(N \times n) \times d}$:

$$O_l, h_l = \text{RNG}(h_{l-1}, X_{E_l}), \tag{4}$$

where the Recurrent Noisy Gating (RNG) is implemented by simple yet effective GRU cells (Dey & Salem, 2017). Then the conditional normal distribution is modeled through the gaussian heads:

$$\mu_l = \text{Linear}_\mu(O_l), \sigma_l = \text{Softplus}(\text{Linear}_\sigma(O_l)), \tag{5}$$
$$\mathbb{P}(\text{R}_l|\text{R}_{1:l-1}) = \mathcal{N}(\mu_l, \sigma_l), \tag{6}$$

where $\mu_l, \sigma_l \in \mathbb{R}^{(N \times n) \times N_r}$, Softplus function is used to keep the standard variance $\sigma_l$ positive, $\mathbb{P}(\text{R}_l|\text{R}_{1:l-1})$ denotes the conditional normal distribution of the routing strategy for $N \times n$ time series tokens in the $l$-th MoE layer. Under this design, RNG-Router can construct the current routing strategy $\text{R}_l$ based on the information from all the previous layers, and adaptively control the degree of retention and forgetting of information from different layers. And the noisy gating mechanism is used to stablize the training of $N_r$ routed experts via resampling from $\mathbb{P}(\text{R}_l|\text{R}_{1:l-1})$:

$$H(X_{E_l}) = \mu_l + \epsilon \odot \sigma_l, \tag{7}$$
$$\text{KeepTopK}(\mathcal{V}, k)_i = \begin{cases} \mathcal{V}_i & \text{if } i \in \text{ArgTopk}(\mathcal{V}) \\ -\infty & \text{otherwise} \end{cases}, \tag{8}$$
$$G(X_{E_l}) = \text{Softmax}(\text{KeepTopK}(H(X_{E_l}), k)), \tag{9}$$

where the Top-$k$ routed experts for each of the $N \times n$ tokens are independently determined through the scores $H(X_{E_l}) \in \mathbb{R}^{(N \times n) \times N_r}$. $\epsilon \in \mathbb{R}^{(N \times n) \times N_r} \sim \mathcal{N}(0, I)$ are used for differentiable resampling. And the gating weights $G(X_{E_l}) \in \mathbb{R}^{(N \times n) \times k}$ of them are calculated through the Softmax function for aggregation of routed experts' outputs. Note that the resampling process shown in Formula (7) only works in the training stage to enhance the roubustness of PatchMoE, and adopts the deterministic values $H(X_{E_l}) = \mu_l$ for inference.

## 3.4 TEMPORAL & CHANNEL LOAD BALANCING LOSS

Since CI-based Transformers may not capture the intricate temporal and channel correlations, we preliminarily handle the bottleneck through simultaneously routing experts for $N \times n$ multivarate time series tokens as aforementioned. To further ensure the sparsification and avoid imbalance in routing, we hope to keep the diversity of routed experts for time series tokens.

As shown in Figure 2 right, the green tokens share distinct groups of routed experts, so that clustering centroids are formed to model the complex correlations. In contrast, red tokens share the same group of experts, which causes imbalance and hinders the representational capability. Intuitively, we design two optimization objectives to encourage the green cases during routing. Specifically, the two optimization objectives consider the relationships between tokens and experts. Take the Channel Load Balancing Loss $\mathcal{L}_{cha}$ in the $l$-th MoE layer as an example:

$$s'_p = \text{reshape}(H(X_{E_l})[:, :, p]) \in \mathbb{R}^{N_r \times N}, \tag{10}$$
$$s^p_{cha} = \text{Softmax}(s'_p) \in \mathbb{R}^{N_r \times N}, \tag{11}$$
$$f_{i,p} = \frac{N_r}{kN} \sum_{t=1}^{N} \mathbf{1}(s^p_{cha}[i, t] \in \text{TopK}(s^p_{cha}[:, t])), \tag{12}$$
$$P_{i,p} = \frac{1}{N} \sum_{t=1}^{N} s^p_{cha}[i, t], \mathcal{L}_{cha} = \sum_{p=1}^{n} \sum_{i=1}^{N_r} f_{i,p} P_{i,p} \tag{13}$$

When calculating the Channel Load Balancing Loss $\mathcal{L}_{cha}$, we parallel along the temporal dimension. $s^p_{cha}[i, t]$ denotes the relationship between $i$-th expert and $t$-th channel of token at the $p$-th temporal index. $F_{i,t,p} = 1$ indicates that the $i$-th expert is one of the TopK routed experts activated for $t$-th channel of token, so that high $f_{i,p}$ indicates that the $i$-th expert is frequenctly activated for all $N$

channel tokens at the $p$-th temporal index, which reflects there exists red cases in routing, causing imbalance. $P_{i,p} \in \mathbb{R}^n$ is the normalization weight. Through weightsuming the channel-wise loss at each time stamp $p$ and then suming up them, the obtained $\mathcal{L}_{cha}$ can measure the degree of imbalance along the channel dimension. Therefore, optimizing $\mathcal{L}_{cha}$ can effectively encourage the modeling of sparse channel correlations, which preserves all tokens of the same channel from sharing the fixed experts, thus keeping load balance.

The Temporal Load Balancing Loss $\mathcal{L}_{tem}$ obeys the same way as Channel Load Balancing Loss. Due to the heterogeneity of temporal patterns, single Feed Forward Layer may not have enough capacity to model these. Through routing tokens from the same channel with distinct groups of experts, the modeling of temporal semantics are boosted. The formulas of Temporal Load Balancing Loss are as follows:

$$s'_t = \text{reshape}(H(X_{E_l})[:,:,t]) \in \mathbb{R}^{N_r \times n}, \tag{14}$$

$$s^t_{tem} = \text{Softmax}(s'_t) \in \mathbb{R}^{N_r \times n}, \tag{15}$$

$$f_{i,t} = \frac{N_r}{kn} \sum_{p=1}^{n} \mathbf{1}(s^t_{tem}[i,p] \in \text{TopK}(s^t_{tem}[:,p])), \tag{16}$$

$$P_{i,t} = \frac{1}{n} \sum_{p=1}^{n} s^t_{tem}[i,p], \mathcal{L}_{tem} = \sum_{t=1}^{N} \sum_{i=1}^{N_r} f_{i,t} P_{i,t} \tag{17}$$

Finally, we integrate the two optimization objectives into the Temporal & Channel Load Balancing Loss $\mathcal{L}_{bal}$:

$$\mathcal{L}_{bal} = \alpha \cdot \mathcal{L}_{tem} + \beta \cdot \mathcal{L}_{cha}, \tag{18}$$

where $\alpha$ and $\beta$ are used to control the sensitivity.

## 3.5 BASIC ARCHITECTURE OF PATCHMOE

Inspired from prior works (Liu et al., 2024a; Riquelme et al., 2021; Ma et al., 2018), PatchMoE replaces the FeedForward Layer in the original Transformers. Instead, each expert in PatchMoE is a FeedForward layer:

$$\text{expert}(X_{E_l}) = \text{Linear}(\text{ReLU}(\text{Linear}(X_{E_l}))) \tag{19}$$

PatchMoE uses $N_r$ finer-grained routed experts and isolates $N_s$ experts as shared ones, where the shared experts model the general patterns and the routed experts are used to model the intricate temporal and channel correlations. Take the $l$-th MoE layer as an example:

$$U = \sum_{i=1}^{N_s} \text{expert}^i_s(X_{E_l}) + \sum_{i=1}^{k} G(X_{E_l})^i \odot \text{expert}^i_r(X_{E_l}), \tag{20}$$

$$V = \text{LayerNorm}(X_{E_l} + U), \tag{21}$$

where $V \in \mathbb{R}^{N \times n \times d}$ is the output of the $l$-th MoE layer, $\text{expert}_s$ denotes the shared experts, $\text{expert}_r$ denotes the routed experts, and $G(X_{E_l})$ is the calculated by RNG-Router to weightsum the routed experts. We make skip connection and adopt LayerNorm to obtain the final output $V$.

## 4 EXPERIMENTS

### 4.1 MAIN RESULTS

#### 4.1.1 EXPERIMENTAL SETTINGS

Since PatchMoE is a cross-task general model for time series analysis, we evaluate it on distinct tasks in an end-to-end manner. For Univariate Forecasting, we evaluate PatchMoE with comprehensive experiments on all the 8,068 univariate time series in TFB (Qiu et al., 2024), and report the Mean Absolute Scaled Error (MASE) and Mean Symmetric Mean Absolute Percentage Error (msMAPE).

For Multivariate Forecasting, we conduct experiments on 8 best-recognized datasets, including ETT (4 subsets), Weather, Electricity, Solar, and Traffic. We follow the protocol in TFB to aviod applying the "Drop Last" trick, adopt Mean Squared Error (MSE) and Mean Absolute Error (MAE) as metrics, and choose the look-back window size in {96, 336, 512} for all datasets and report each method's best results.

For Anomaly Detection, we conduct experiments using 8 real-world datasets from TAB (Qiu et al., 2025a). We report the results on datasets including CalIT2, Credit, GECCO, Genesis, MSL, NYC, PSM, and SMD, adopting the Label-based metric Affiliated-F1-score (F), and Score-based metric: Area under the Receiver Operating Characteristics Curve (AUC) as main evaluation metrics.

For Imputation, we use datasets from electricity and weather domains, selecting ETT (4 subsets), Electricity, and Weather as benchmarks, and report Mean Squared Error (MSE) and Mean Absolute Error (MAE) as main metrics. We adopt four mask ratios (randomly masking) {12.5%, 25%, 37.5%, 50%} with the input length equals 1,024 on each dataset, and report the average performance.

Time series classification can be used in medical diagnosis and recognition. To evaluate the sequence-level classification capability of PatchMoE, we choose 10 datasets from UEA Time Series Classification Archive (Bagnall et al., 2018) and report the average accuracy of each model.

### 4.1.2 BASELINES

Our baselines include task-agnostic models like iTransformer (Liu et al., 2024c), PatchTST (Nie et al., 2023), Crossformer (Zhang & Yan, 2022), TimesNet (Wu et al., 2023), DLinear (Zeng et al., 2023), and FEDformer (Zhou et al., 2022a), and task-specific models like Flowformer (Wu et al., 2022), LighTS (Zhang et al., 2022). CATCH (Wu et al., 2025b), DCdetector (Yang et al., 2023), Anomaly Transformer (Xu et al., 2021), Rocket (Dempster et al., 2020), and MoE-based models, i.e., Pathformer (Chen et al., 2024) and Time-MoE (Full-shot) (Shi et al., 2024).

### 4.1.3 UNIVARIATE FORECASTING

As shown in Figure 3, PatchMoE achieves the best performance on the 8,068 datasets. Compared with previous advanced models Times-Net and PatchTST, PatchMoE shows more stable performance with lower average msMAPE values. Compared with recent strong models like Amplifier and TimeKAN, PatchMoE also achieves 5.2% and 3.9% reduction on msMAPE, demonstrating the state-of-the-art performance.

Figure 3: Model comparison in univariate forecasting. The msMAPE results are average from 8,068 univariate time series in TFB (lower is better). See Table 9 in Appendix B for full results.

### 4.1.4 MULTIVARIATE FORECASTING

As shown in Table 1, PatchMoE consistently outperforms other models across various datasets. Compared with PatchTST, Patch-MoE's mixture-of-experts mechanism introduces consistent improvement on all datasets, demonstrating stronger representational capability. Considering large datasets, PatchMoE possesses 7.6% lower MSE and 7.0% lower MAE on Electricity, 9.0% lower MSE and 21.8% lower MAE on Solar, demonstrating the larger model capacity on these large datasets. Compared with CD-based models like Crossformer and iTransformer, PatchMoE also has better performance on datasets with significant channel correlations (like Traffic and Solar), demonstrating the effectiveness of the Routing strategy and the Temporal & Channel Load Balancing Loss. Note that PatchMoE patchifys the multivariate time series in a CI manner but can capture the token-wise channel correlations.

### 4.1.5 ANOMALY DETECTION

The results are listed in Table 2. Compared with advanced approaches, it can be seen that PatchMoE achieves SOTA results under the widely used Affiliated-F1-score and AUC-ROC metrics in most

Table 1: Multivariate forecasting average results with forecasting horizons $F \in \{96, 192, 336, 720\}$ for the datasets. Lower Mean Squared Error (MSE) and Mean Absolute Error (MAE) values indicate better performance. **Bond**: the best, Underline: the 2nd best. Full results are available in Table 10 of Appendix B. For Time-MoE, Electricity, Solar and Traffic are included in pretraining datasets.

| Datasets | ETTh1 | | ETTh2 | | ETTm1 | | ETTm2 | | Weather | | Electricity | | Solar | | Traffic | |
|---|---|---|---|---|---|---|---|---|---|---|---|---|---|---|---|---|
| Metrics | MSE | MAE | MSE | MAE | MSE | MAE | MSE | MAE | MSE | MAE | MSE | MAE | MSE | MAE | MSE | MAE |
| FEDformer [2022] | 0.433 | 0.454 | 0.406 | 0.438 | 0.567 | 0.519 | 0.335 | 0.380 | 0.312 | 0.356 | 0.219 | 0.330 | 0.641 | 0.628 | 0.620 | 0.382 |
| DLinear [2023] | 0.430 | 0.443 | 0.470 | 0.468 | 0.356 | 0.378 | 0.259 | 0.324 | 0.242 | 0.295 | 0.167 | 0.264 | 0.224 | 0.286 | 0.418 | 0.287 |
| TimesNet [2023] | 0.468 | 0.459 | 0.390 | 0.417 | 0.408 | 0.415 | 0.292 | 0.331 | 0.255 | 0.282 | 0.190 | 0.284 | 0.211 | 0.281 | 0.617 | 0.327 |
| Crossformer [2023] | 0.439 | 0.461 | 0.894 | 0.680 | 0.464 | 0.456 | 0.501 | 0.505 | 0.232 | 0.294 | 0.171 | 0.263 | 0.205 | 0.232 | 0.522 | 0.282 |
| PatchTST [2023] | 0.419 | 0.436 | 0.351 | 0.395 | 0.349 | 0.381 | 0.256 | 0.314 | 0.224 | 0.262 | 0.171 | 0.270 | 0.200 | 0.284 | 0.397 | 0.275 |
| TimeMixer [2024] | 0.427 | 0.441 | 0.347 | 0.394 | 0.356 | 0.380 | 0.257 | 0.318 | 0.225 | 0.263 | 0.185 | 0.284 | 0.203 | 0.261 | 0.410 | 0.279 |
| Pathformer [2024] | 0.417 | 0.426 | 0.360 | 0.395 | 0.357 | 0.375 | 0.309 | 0.250 | 0.227 | 0.263 | 0.160 | 0.253 | 0.204 | 0.230 | 0.418 | 0.281 |
| iTransformer [2024] | 0.440 | 0.445 | 0.359 | 0.396 | 0.347 | 0.378 | 0.258 | 0.318 | 0.232 | 0.270 | 0.163 | 0.258 | 0.202 | 0.260 | 0.397 | 0.281 |
| Amplifier [2025] | 0.421 | 0.433 | 0.356 | 0.402 | 0.353 | 0.379 | 0.256 | 0.318 | 0.223 | 0.264 | 0.163 | 0.256 | 0.202 | 0.256 | 0.417 | 0.290 |
| TimeKAN [2025] | 0.409 | 0.427 | 0.350 | 0.397 | 0.344 | 0.380 | 0.260 | 0.318 | 0.226 | 0.268 | 0.164 | 0.258 | 0.198 | 0.263 | 0.420 | 0.286 |
| Time-MoE [2025] | **0.379** | **0.406** | 0.346 | 0.386 | 0.345 | 0.381 | 0.271 | 0.335 | 0.236 | 0.275 | - | - | - | - | - | - |
| PatchMoE [ours] | 0.400 | 0.424 | **0.340** | **0.384** | **0.343** | **0.370** | **0.251** | **0.306** | **0.221** | **0.250** | **0.158** | **0.251** | **0.182** | **0.222** | **0.392** | **0.274** |

benchmark datasets. It mean that PatchMoE possesses stable performance under different anomaly thresholds, which is highly important for real-world applications. Compared with the most advanced baseline CATCH (Wu et al., 2025b), PatchMoE also shows higher accuracy and considers patch-wise fine-grained channel correlations in a more lightweight manner on some cases.

Table 2: Anomaly detection results. Higher Affiliated-F1 (F) and AUC-ROC (AUC) values indicate better performance. **Bond**: the best, Underline: the 2nd best. Full results are available in Table 12 of Appendix B.

| Datasets | CalIt2 | | Credit | | GECCO | | Genesis | | MSL | | NYC | | PSM | | SMD | |
|---|---|---|---|---|---|---|---|---|---|---|---|---|---|---|---|---|
| Metrics | F | AUC | F | AUC | F | AUC | F | AUC | F | AUC | F | AUC | F | AUC | F | AUC |
| ATransformer [2022] | 0.688 | 0.491 | 0.646 | 0.533 | 0.782 | 0.516 | 0.715 | 0.472 | 0.685 | 0.508 | 0.691 | 0.499 | 0.654 | 0.498 | 0.704 | 0.309 |
| FEDformer [2022] | 0.788 | 0.707 | 0.683 | 0.825 | 0.900 | 0.709 | 0.893 | 0.802 | 0.726 | 0.561 | 0.691 | 0.725 | 0.761 | **0.679** | 0.782 | 0.650 |
| DCdetector [2023] | 0.673 | 0.527 | 0.610 | 0.504 | 0.671 | 0.555 | 0.776 | 0.507 | 0.683 | 0.504 | 0.698 | 0.528 | 0.662 | 0.499 | 0.675 | 0.500 |
| DLinear [2023] | 0.793 | 0.752 | 0.738 | 0.954 | 0.893 | 0.947 | 0.856 | 0.696 | 0.725 | 0.624 | 0.828 | 0.768 | 0.831 | 0.580 | 0.841 | 0.728 |
| TimesNet [2023] | 0.794 | 0.771 | 0.744 | 0.958 | 0.897 | 0.964 | 0.864 | 0.913 | 0.734 | 0.613 | 0.794 | 0.791 | 0.842 | 0.592 | 0.833 | 0.766 |
| Crossformer [2023] | 0.789 | 0.798 | 0.720 | 0.951 | 0.897 | 0.770 | 0.865 | 0.755 | 0.733 | 0.587 | 0.692 | 0.679 | 0.789 | 0.654 | 0.839 | 0.710 |
| PatchTST [2023] | 0.660 | 0.808 | 0.746 | 0.957 | 0.906 | 0.949 | 0.856 | 0.685 | 0.723 | 0.637 | 0.776 | 0.709 | 0.831 | 0.586 | 0.845 | 0.736 |
| ModernTCN [2024] | 0.780 | 0.676 | 0.744 | 0.957 | 0.899 | 0.954 | 0.833 | 0.676 | 0.726 | 0.633 | 0.769 | 0.466 | 0.825 | 0.592 | 0.840 | 0.722 |
| iTransformer [2024] | 0.812 | 0.791 | 0.713 | 0.934 | 0.839 | 0.794 | 0.891 | 0.690 | 0.710 | 0.611 | 0.684 | 0.640 | 0.853 | 0.592 | 0.827 | 0.745 |
| CATCH [2025] | 0.835 | 0.838 | 0.750 | 0.958 | 0.908 | 0.970 | 0.896 | **0.974** | 0.740 | **0.664** | **0.994** | 0.816 | **0.859** | 0.652 | 0.847 | 0.811 |
| PatchMoE [ours] | **0.842** | **0.861** | **0.754** | **0.959** | **0.914** | **0.979** | **0.903** | 0.862 | **0.746** | 0.641 | 0.973 | **0.833** | 0.850 | 0.645 | **0.868** | **0.831** |

### 4.1.6 IMPUTATION

Table 3 presents PatchMoE's performance in imputating missing values. We observe that Patch-MoE consistently outperforms all baselines, demonstrating its potential of being the infrastructure for data preprocessing in real-world applications. Compared with the most advanced baseline TimeMixer++ (Wang et al., 2024), PatchMoE surpasses it significantly on the Electricity and Weather datasets, showing the excellent model capacity for large datasets.

### 4.1.7 CLASSIFICATION

See Figure 4, PatchMoE demonstrates remarkable capabilities in time series classification. Compared with generative tasks like forecasting, anomaly detection, and imputation, classification is a discriminative task which relies more on model's sequence-aware capability and channel correlations. Our proposed PatchMoE can learn the overall characteristics of a time series via modeling the local patch-wise transition rule, and capture the intricate channel correlations through the routing strategy, thus it achieves the state-of-the-art performance on classification tasks.

Table 3: Multivariate imputation average results with mask ratios spanning $\{12.5\%, 25\%, 37.5\%, 50\%\}$ for the datasets. **Bond**: the best, Underline: the 2nd best.

| Datasets | ETT (Avg) | | Electricity | | Weather | |
|---|---|---|---|---|---|---|
| Metrics | MSE | MAE | MSE | MAE | MSE | MAE |
| Autoformer [2022] | 0.104 | 0.215 | 0.141 | 0.234 | 0.066 | 0.107 |
| FEDformer [2022] | 0.124 | 0.230 | 0.181 | 0.314 | 0.064 | 0.139 |
| MICN [2023] | 0.119 | 0.234 | 0.138 | 0.246 | 0.075 | 0.126 |
| TimesNet [2023] | 0.079 | 0.182 | 0.135 | 0.255 | 0.061 | 0.098 |
| DLinear [2023] | 0.115 | 0.229 | 0.080 | 0.200 | 0.071 | 0.107 |
| TIDE [2023] | 0.314 | 0.366 | 0.182 | 0.202 | 0.063 | 0.131 |
| Crossformer [2023] | 0.150 | 0.258 | 0.125 | 0.204 | 0.150 | 0.111 |
| PatchTST [2023] | 0.120 | 0.225 | 0.129 | 0.198 | 0.082 | 0.149 |
| iTransformer [2024] | 0.096 | 0.205 | 0.140 | 0.223 | 0.095 | 0.102 |
| TimeMixer [2024] | 0.097 | 0.220 | 0.142 | 0.261 | 0.091 | 0.114 |
| TimeMixer++ [2025] | 0.055 | 0.154 | 0.109 | 0.197 | 0.049 | 0.078 |
| PatchMoE [ours] | **0.054** | **0.154** | **0.052** | **0.162** | **0.035** | **0.064** |

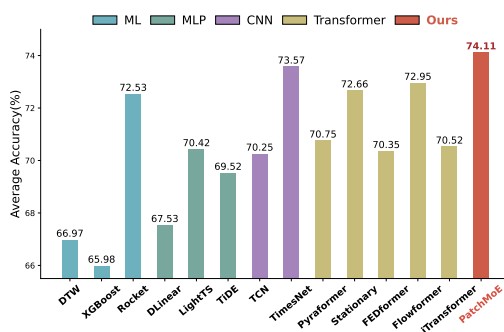

Figure 4: Model comparison in classification. The accuracy are averaged from 10 subsets from UEA. See Table 11 in Appendix B for full results.

## 4.2 MODEL ANALYSIS

### 4.2.1 ABLATION STUDIES

To verify the effectiveness of PatchMoE, we conduct ablation studies on the components different from traditional MoE architectures, i.e., RNG-Router, Shared Experts, and Temporal & Channel Load Balancing Loss. The results are shown in Table 4, PatchMoE with all above components achieves the best performance. The RNG-Router plays the most critical role to consider the hierarchical representation differences in routing, improving the performance by reducing 4.2% in MSE. The Shared Experts are crucial on large datasets like Traffic, which can enhance the model capacity to effectively capture the general patterns, lead-

Table 4: Studies on key components of Patch-MoE, inlcuding w/o RNG-Router (line 1), w/o Shared Experts (line 2), w/o Temporal & Channel Load Balancing Loss (line 3), and original Patch-MoE (line 4). Full results are in Appendix 13.

| ETTh1 | | ETTm2 | | Solar | | Traffic | |
|---|---|---|---|---|---|---|---|
| MSE | MAE | MSE | MAE | MSE | MAE | MSE | MAE |
| 0.417 | 0.434 | 0.260 | 0.316 | 0.197 | 0.228 | 0.398 | 0.276 |
| 0.412 | 0.434 | 0.257 | 0.313 | 0.188 | 0.228 | 0.421 | 0.293 |
| 0.403 | 0.426 | 0.257 | 0.311 | 0.185 | 0.226 | 0.403 | 0.285 |
| **0.400** | **0.424** | **0.251** | **0.306** | **0.182** | **0.222** | **0.392** | **0.274** |

ing 6.9% reduction in MSE. The Temporal & Channel Load Balancing Loss boosts the clustering of correlated temporal- and channel-wise tokens, consistently enhancing the performance.

## 4.3 MORE ANALYTICS

**Parameter Sensitivity**. We study the parameter sensitivity of PatchMoE–see Figure 7 in Appendix B. PatchMoE achieves strong performance under the parameter configurations of patch size $p = 24$, number of hidden layers $L = 3$, and number of routed experts $N^r = 10$.

**Representation Analytics**. We provide the representation analytics in Figure 6 in Appendix B. Results demonstarte that RNG-Router can effectively utilize the hierarchical representations to boost the routing of time series tokens for distinct tasks, thus possessing task-specific capabilities.

## 5 CONCLUSION

In this paper, we propose a general representation learning framework, called PatchMoE, with a novel Mixture-of-Experts architecture tailored for time series analysis. To sum up, PatchMoE can utilize the hierarchical representation differences across different neural layers via a RNG-Router, making accurate routing decision based on the current task. And the Temporal & Channel Load Balancing Loss is devised to encourage the modeling of sparse correlations. PatchMoE also utilizes the shared experts to capture common patterns and routed experts to capture detailed differences. Based on these innovative mechanisms, PatchMoE demonstrates state-of-the-art performances on time series analytics.

ETHICS STATEMENT

Our work exclusively uses publicly available benchmark datasets that contain no personally identifiable information. No human subjects are involved in this research.

REPRODUCIBILITY STATEMENT

We promise that all experimental results can be reproduced. We have released our model code in an anonymous repository: https://anonymous.4open.science/r/PatchMoE-BD38.

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

## THE USE OF LARGE LANGUAGE MODELS (LLMS)

We do not use Large Language Models in our methodology and writing.

## A    IMPLEMENTATION DETAILS

We introduce the Dataset Details, Metric Details, and Experimental Details in this section for clarity.

### A.1    DATASET DETAILS

We evaluate the performance of different models for multivariate forecasting on 8 well-established datasets from TFB, including Weather, Traffic, Electricity, Solar, and ETT datasets (ETTh1, ETTh2, ETTm1, ETTm2), and provide their detailed descriptions in Table 6. For univariate forecasting, we evaluate all 8,068 well-established univariate time series from TFB, as summarized in Table 5. For anomaly detection, we evaluate 9 well-established datasets from TAB, including CalIt2, Credit, GECCO, Genesis, MSL, NYC, PSM, SMAP, and SMD, with detailed descriptions provided in Table 7. We evaluate 10 datasets from the UEA Time Series Classification Archive for classification, and show their details in Table 8. For imputation, we evaluate the Electricity, Weather, and ETT datasets (ETTh1, ETTh2, ETTm1, ETTm2).

Table 5: Univariate forecasting dataset detailed descriptions.

| Dataset | Series Count | Input | Predict | Avg Length | Frequency |
|---|---|---|---|---|---|
| TFB-Yearly | 1,500 | 7 | 6 | 32.0 | yearly |
| TFB-Quarterly | 1,514 | 10 | 8 | 97.2 | quarterly |
| TFB-Monthly | 1,674 | 22 | 18 | 259.1 | monthly |
| TFB-Weekly | 805 | 16 | 13 | 536.3 | weekly |
| TFB-Daily | 1,484 | 17 | 14 | 4,950.8 | daily |
| TFB-Hourly | 706 | 60 | 48 | 5,109.0 | hourly |
| TFB-Other | 385 | 10 | 8 | 1,678.4 | other |

Table 6: Multivariate forecasting dataset detailed descriptions (Split: Train/Validation/Test split ratio).

| Dataset | Dim | Input | Predict | Length | Frequency | Split | Domain |
|---|---|---|---|---|---|---|---|
| ETTm1 | 7 | {96, 336, 512} | {96, 192, 336, 720} | 57,600 | 15min | 6:2:2 | Electricity |
| ETTm2 | 7 | {96, 336, 512} | {96, 192, 336, 720} | 57,600 | 15min | 6:2:2 | Electricity |
| ETTh1 | 7 | {96, 336, 512} | {96, 192, 336, 720} | 14,400 | 15 min | 6:2:2 | Electricity |
| ETTh2 | 7 | {96, 336, 512} | {96, 192, 336, 720} | 14,400 | 15 min | 6:2:2 | Electricity |
| Electricity | 321 | {96, 336, 512} | {96, 192, 336, 720} | 26,304 | Hourly | 7:1:2 | Electricity |
| Traffic | 862 | {96, 336, 512} | {96, 192, 336, 720} | 17,544 | Hourly | 7:1:2 | Traffic |
| Weather | 21 | {96, 336, 512} | {96, 192, 336, 720} | 52,696 | 10 min | 7:1:2 | Environment |
| Solar | 137 | {96, 336, 512} | {96, 192, 336, 720} | 52,560 | 10min | 6:2:2 | Energy |

Table 7: Anomaly detection dataset detailed descriptionss (AR: anomaly ratio).

| Dataset | Dim | AR(%) | Length | Test Length | Domain |
|---|---|---|---|---|---|
| CalIt2 | 2 | 4.09 | 5,040 | 2,520 | Visitors Flowrate |
| GECCO | 9 | 1.25 | 138,521 | 69,261 | Water Treatment |
| Credit | 29 | 0.17 | 284,807 | 142,404 | Finance |
| Genesis | 18 | 0.31 | 16,220 | 12,616 | Machinery |
| NYC | 3 | 0.57 | 17,520 | 4,416 | Transport |
| MSL | 55 | 5.88 | 132,046 | 73,729 | Spacecraft |
| SMAP | 25 | 9.72 | 562,800 | 427,617 | Spacecraft |
| PSM | 25 | 11.07 | 220,322 | 87,841 | Server Machine |
| SMD | 38 | 2.08 | 1,416,825 | 708,420 | Server Machine |

Table 8: Classification dataset detailed descriptions.

| Dataset | Dim | Train Cases | Test Cases | Series Length | Classes |
|---|---|---|---|---|---|
| EthanolConcentration | 3 | 261 | 263 | 1,751 | 4 |
| FaceDetection | 144 | 5,890 | 3,524 | 62 | 2 |
| Handwriting | 3 | 150 | 850 | 152 | 26 |
| Heartbeat | 61 | 204 | 205 | 405 | 2 |
| JapaneseVowels | 12 | 270 | 370 | 29 | 9 |
| PEMS-SF | 963 | 267 | 173 | 144 | 7 |
| SelfRegulationSCP1 | 6 | 268 | 293 | 896 | 2 |
| SelfRegulationSCP2 | 7 | 200 | 180 | 1,152 | 2 |
| SpokenArabicDigits | 13 | 6,599 | 2,199 | 93 | 10 |
| UWaveGestureLibrary | 3 | 120 | 320 | 315 | 8 |

### A.2    EXPERIMENTAL DETAILS

All experiments are conduct using PyTorch and executed on an NVIDIA Tesla-A800 GPU. The training process is guided by the L1 or L2 loss, and optimized with the ADAM optimizer. The "Drop Last" tricky is forbidden. We conduct 8 sets of hyperparameter search for each baseline and PatchMoE and save their best parameters. For the best parameter, we run it 5 times with different random seeds and report the mean values.

### A.3    METRIC DETAILS

Regarding evaluation metrics, following the experimental setup in TFB, we adopt Mean Squared Error (MSE) and Mean Absolute Error (MAE) as evaluation metrics for multivariate forecasting. For univariate forecasting, we use Modified Symmetric Mean Absolute Percentage Error (MSMAPE) and Mean Absolute Scaled Error (MASE). $M$ is the length of the training series, $S$ is the seasonality of the time series, $h$ is the forecasting horizon, the $F_k$ are the generated forecasts, and the $Y_k$ are

the actual values. We set parameter $\epsilon$ in Equation 25 to its proposed default of 0.1. For rolling forecasting, we further calculate the average of error metrics for all samples (windows) on each time series to assess method performance. The definitions of these metrics are as follows:

$$MSE = \frac{1}{h}\sum_{k=1}^{h}(F_k - Y_k)^2, \tag{22}$$

$$MAE = \frac{1}{h}\sum_{k=1}^{h}|F_k - Y_k|, \tag{23}$$

$$MASE = \frac{\sum_{k=M+1}^{M+h}|F_k - Y_k|}{\frac{h}{M-S}\sum_{k=S+1}^{M}|Y_k - Y_{k-S}|}, \tag{24}$$

$$MSMAPE = \frac{100\%}{h}\sum_{k=1}^{h}\frac{|F_k - Y_k|}{\max(|Y_k| + |F_k| + \epsilon, 0.5 + \epsilon)/2}, \tag{25}$$

## B  FULL RESULTS

We list the full results in this section–see Table 9-12, including Univariate Forecasting, Multivariate Forecasting, Anomaly Detection, and Classification. In summary, PatchMoE achieves consistent state-of-the-art performance on all five tasks–see Figure 5.

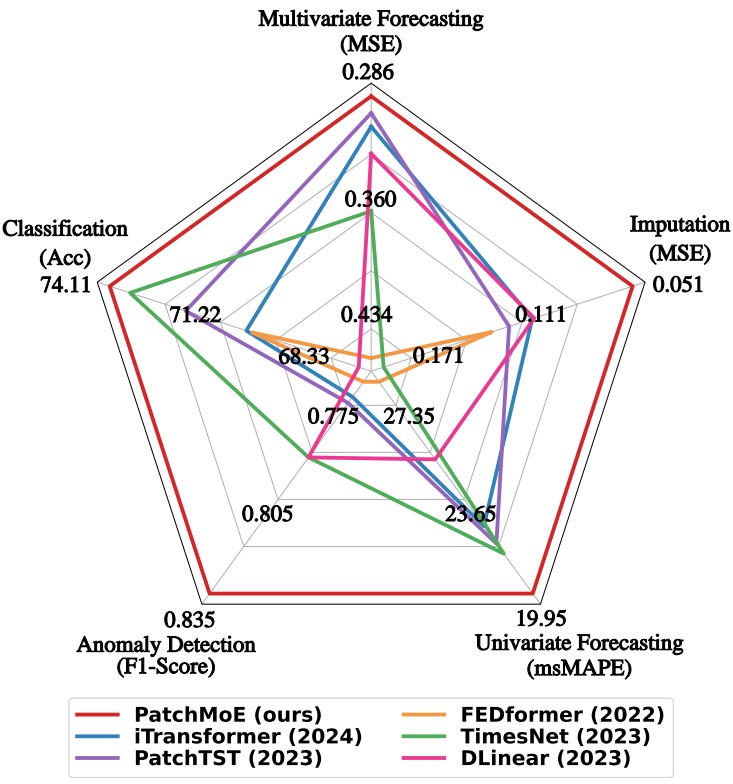

Figure 5: Model Performance comparision in five tasks.

Table 9: Univariate forecasting results averaged over 8,068 time series from TFB. Lower msMAPE and MASE values indicate better performance. **Red**: the best, **Blue**: the 2nd best.

| Models | PatchMoE (ours) | TimeKAN (2025) | Ampilifier (2025) | iTransformer (2024) | TimeMixer (2024) | PatchTST (2023) | Crossformer (2023) | TimesNet (2023) | DLinear (2023) | N-HITS (2023) | Stationary (2022) | FEDformer (2022) | N-BEATS (2020) | TCN (2018) | LR (2005) | RF (2001) |
|---|---|---|---|---|---|---|---|---|---|---|---|---|---|---|---|---|
| msMAPE | **19.95** | 20.77 | 21.05 | 22.53 | 21.02 | 21.87 | 176.57 | 21.48 | 25.09 | 24.79 | 21.77 | 28.06 | 26.93 | 132.47 | 29.79 | 22.82 |
| MASE | **1.97** | 2.23 | 2.02 | 2.59 | 2.16 | 2.35 | 29.22 | 2.34 | 2.67 | 2.55 | 2.35 | 2.79 | 2.64 | 18.27 | 4.44 | 2.41 |

Table 10: Multivariate forecasting results with forecasting horizons $F \in \{96, 192, 336, 720\}$ for the datasets. Lower Mean Squared Error (MSE ) and Mean Absolute Error (MAE) values indicate better performance. **Red**: the best, Blue: the 2nd best.

| Models | | PatchMoE (ours) | | Time-MoE (2025) | | TimeKAN (2025) | | Amplifier (2025) | | iTransformer (2024) | | Pathformer (2024) | | TimeMixer (2023) | | PatchTST (2023) | | Crossformer (2023) | | TimesNet (2023) | | DLinear (2022) | | FEDformer (2022) | |
|---|---|---|---|---|---|---|---|---|---|---|---|---|---|---|---|---|---|---|---|---|---|---|---|---|---|
| Metrics | | MSE | MAE | MSE | MAE | MSE | MAE | MSE | MAE | MSE | MAE | MSE | MAE | MSE | MAE | MSE | MAE | MSE | MAE | MSE | MAE | MSE | MAE | MSE | MAE |
| ETTh1 | 96 | 0.355 | 0.390 | 0.345 | 0.373 | 0.370 | 0.396 | 0.373 | 0.399 | 0.386 | 0.405 | 0.372 | 0.392 | 0.372 | 0.401 | 0.377 | 0.397 | 0.411 | 0.435 | 0.389 | 0.412 | 0.379 | 0.403 | 0.379 | 0.419 |
| | 192 | 0.398 | 0.417 | 0.372 | 0.396 | 0.403 | 0.417 | 0.414 | 0.420 | 0.430 | 0.435 | 0.408 | 0.415 | 0.413 | 0.430 | 0.409 | 0.425 | 0.409 | 0.438 | 0.440 | 0.443 | 0.427 | 0.435 | 0.420 | 0.444 |
| | 336 | 0.418 | 0.431 | 0.389 | 0.412 | 0.420 | 0.432 | 0.442 | 0.446 | 0.450 | 0.452 | 0.438 | 0.434 | 0.438 | 0.450 | 0.431 | 0.444 | 0.433 | 0.457 | 0.523 | 0.487 | 0.440 | 0.440 | 0.458 | 0.466 |
| | 720 | 0.430 | 0.456 | 0.410 | 0.443 | 0.442 | 0.463 | 0.455 | 0.467 | 0.495 | 0.487 | 0.450 | 0.463 | 0.483 | 0.483 | 0.457 | 0.477 | 0.501 | 0.514 | 0.521 | 0.495 | 0.473 | 0.494 | 0.474 | 0.488 |
| ETTh2 | 96 | 0.272 | 0.330 | 0.276 | 0.340 | 0.280 | 0.343 | 0.287 | 0.349 | 0.292 | 0.347 | 0.279 | 0.336 | 0.270 | 0.342 | 0.274 | 0.337 | 0.728 | 0.603 | 0.334 | 0.370 | 0.300 | 0.364 | 0.337 | 0.380 |
| | 192 | 0.333 | 0.376 | 0.331 | 0.371 | 0.329 | 0.382 | 0.348 | 0.393 | 0.348 | 0.384 | 0.345 | 0.380 | 0.349 | 0.387 | 0.348 | 0.384 | 0.723 | 0.607 | 0.404 | 0.413 | 0.387 | 0.423 | 0.415 | 0.428 |
| | 336 | 0.357 | 0.398 | 0.373 | 0.402 | 0.370 | 0.412 | 0.383 | 0.423 | 0.372 | 0.407 | 0.378 | 0.408 | 0.367 | 0.410 | 0.377 | 0.416 | 0.740 | 0.628 | 0.389 | 0.435 | 0.490 | 0.487 | 0.389 | 0.457 |
| | 720 | 0.396 | 0.433 | 0.404 | 0.431 | 0.420 | 0.450 | 0.407 | 0.444 | 0.424 | 0.444 | 0.437 | 0.455 | 0.401 | 0.436 | 0.406 | 0.441 | 1.386 | 0.882 | 0.434 | 0.448 | 0.704 | 0.597 | 0.483 | 0.488 |
| ETTm1 | 96 | 0.282 | 0.332 | 0.286 | 0.334 | 0.290 | 0.348 | 0.292 | 0.346 | 0.287 | 0.342 | 0.290 | 0.335 | 0.293 | 0.345 | 0.289 | 0.343 | 0.314 | 0.367 | 0.340 | 0.378 | 0.300 | 0.345 | 0.463 | 0.463 |
| | 192 | 0.325 | 0.357 | 0.307 | 0.358 | 0.332 | 0.368 | 0.327 | 0.365 | 0.331 | 0.371 | 0.337 | 0.363 | 0.335 | 0.372 | 0.329 | 0.368 | 0.374 | 0.410 | 0.392 | 0.404 | 0.336 | 0.366 | 0.575 | 0.516 |
| | 336 | 0.359 | 0.379 | 0.354 | 0.390 | 0.354 | 0.386 | 0.365 | 0.386 | 0.358 | 0.384 | 0.374 | 0.384 | 0.368 | 0.386 | 0.362 | 0.390 | 0.413 | 0.432 | 0.423 | 0.426 | 0.367 | 0.386 | 0.618 | 0.544 |
| | 720 | 0.407 | 0.412 | 0.433 | 0.445 | 0.401 | 0.417 | 0.427 | 0.419 | 0.412 | 0.416 | 0.428 | 0.416 | 0.426 | 0.417 | 0.416 | 0.423 | 0.753 | 0.613 | 0.475 | 0.453 | 0.419 | 0.416 | 0.612 | 0.551 |
| ETTm2 | 96 | 0.160 | 0.244 | 0.172 | 0.265 | 0.164 | 0.254 | 0.164 | 0.254 | 0.168 | 0.262 | 0.164 | 0.250 | 0.165 | 0.256 | 0.165 | 0.255 | 0.296 | 0.391 | 0.189 | 0.265 | 0.164 | 0.255 | 0.216 | 0.309 |
| | 192 | 0.217 | 0.285 | 0.228 | 0.306 | 0.238 | 0.300 | 0.226 | 0.300 | 0.224 | 0.295 | 0.219 | 0.288 | 0.225 | 0.298 | 0.221 | 0.293 | 0.369 | 0.416 | 0.254 | 0.310 | 0.224 | 0.304 | 0.297 | 0.360 |
| | 336 | 0.273 | 0.322 | 0.281 | 0.345 | 0.278 | 0.331 | 0.276 | 0.331 | 0.274 | 0.330 | 0.267 | 0.319 | 0.277 | 0.332 | 0.276 | 0.327 | 0.588 | 0.600 | 0.313 | 0.345 | 0.277 | 0.337 | 0.400 | 0.400 |
| | 720 | 0.355 | 0.373 | 0.403 | 0.424 | 0.359 | 0.387 | 0.358 | 0.388 | 0.367 | 0.385 | 0.361 | 0.377 | 0.360 | 0.387 | 0.362 | 0.381 | 0.750 | 0.612 | 0.413 | 0.402 | 0.371 | 0.401 | 0.459 | 0.450 |
| Weather | 96 | 0.145 | 0.183 | 0.151 | 0.203 | 0.151 | 0.202 | 0.147 | 0.199 | 0.157 | 0.207 | 0.148 | 0.195 | 0.147 | 0.198 | 0.150 | 0.200 | 0.143 | 0.206 | 0.168 | 0.214 | 0.170 | 0.230 | 0.229 | 0.298 |
| | 192 | 0.190 | 0.228 | 0.195 | 0.246 | 0.195 | 0.244 | 0.194 | 0.245 | 0.200 | 0.248 | 0.191 | 0.235 | 0.191 | 0.242 | 0.191 | 0.239 | 0.195 | 0.261 | 0.219 | 0.262 | 0.216 | 0.275 | 0.265 | 0.334 |
| | 336 | 0.240 | 0.269 | 0.247 | 0.288 | 0.242 | 0.288 | 0.243 | 0.282 | 0.252 | 0.287 | 0.243 | 0.274 | 0.244 | 0.280 | 0.242 | 0.279 | 0.254 | 0.319 | 0.278 | 0.302 | 0.258 | 0.307 | 0.330 | 0.372 |
| | 720 | 0.309 | 0.321 | 0.352 | 0.366 | 0.317 | 0.340 | 0.310 | 0.329 | 0.320 | 0.336 | 0.318 | 0.326 | 0.316 | 0.331 | 0.312 | 0.330 | 0.335 | 0.385 | 0.353 | 0.351 | 0.324 | 0.367 | 0.423 | 0.418 |
| Electricity | 96 | 0.131 | 0.226 | - | - | 0.135 | 0.231 | 0.132 | 0.227 | 0.134 | 0.230 | 0.135 | 0.222 | 0.153 | 0.256 | 0.143 | 0.247 | 0.134 | 0.231 | 0.169 | 0.271 | 0.140 | 0.237 | 0.191 | 0.305 |
| | 192 | 0.145 | 0.240 | - | - | 0.149 | 0.243 | 0.149 | 0.243 | 0.154 | 0.250 | 0.157 | 0.253 | 0.168 | 0.269 | 0.158 | 0.260 | 0.146 | 0.243 | 0.180 | 0.280 | 0.154 | 0.250 | 0.203 | 0.316 |
| | 336 | 0.162 | 0.256 | - | - | 0.165 | 0.260 | 0.167 | 0.261 | 0.169 | 0.265 | 0.170 | 0.267 | 0.189 | 0.291 | 0.168 | 0.267 | 0.165 | 0.264 | 0.204 | 0.293 | 0.169 | 0.268 | 0.221 | 0.333 |
| | 720 | 0.193 | 0.282 | - | - | 0.206 | 0.297 | 0.203 | 0.292 | 0.194 | 0.288 | 0.211 | 0.302 | 0.228 | 0.320 | 0.214 | 0.307 | 0.237 | 0.314 | 0.206 | 0.293 | 0.203 | 0.300 | 0.259 | 0.364 |
| Solar | 96 | 0.166 | 0.207 | - | - | 0.187 | 0.255 | 0.175 | 0.237 | 0.174 | 0.229 | 0.218 | 0.235 | 0.180 | 0.233 | 0.170 | 0.234 | 0.183 | 0.208 | 0.198 | 0.270 | 0.199 | 0.265 | 0.485 | 0.570 |
| | 192 | 0.178 | 0.222 | - | - | 0.194 | 0.265 | 0.198 | 0.259 | 0.205 | 0.270 | 0.196 | 0.220 | 0.201 | 0.259 | 0.204 | 0.302 | 0.208 | 0.226 | 0.206 | 0.276 | 0.228 | 0.282 | 0.415 | 0.477 |
| | 336 | 0.184 | 0.224 | - | - | 0.203 | 0.264 | 0.213 | 0.259 | 0.216 | 0.282 | 0.195 | 0.228 | 0.214 | 0.272 | 0.212 | 0.293 | 0.212 | 0.239 | 0.208 | 0.284 | 0.234 | 0.295 | 1.008 | 0.839 |
| | 720 | 0.198 | 0.234 | - | - | 0.209 | 0.269 | 0.222 | 0.269 | 0.211 | 0.260 | 0.208 | 0.237 | 0.218 | 0.278 | 0.215 | 0.307 | 0.215 | 0.256 | 0.232 | 0.294 | 0.243 | 0.301 | 0.655 | 0.627 |
| Traffic | 96 | 0.361 | 0.261 | - | - | 0.388 | 0.269 | 0.391 | 0.277 | 0.363 | 0.265 | 0.392 | 0.271 | 0.369 | 0.256 | 0.370 | 0.262 | 0.526 | 0.288 | 0.595 | 0.312 | 0.395 | 0.275 | 0.593 | 0.365 |
| | 192 | 0.382 | 0.268 | - | - | 0.411 | 0.286 | 0.405 | 0.283 | 0.384 | 0.273 | 0.405 | 0.274 | 0.400 | 0.271 | 0.386 | 0.269 | 0.503 | 0.263 | 0.613 | 0.322 | 0.407 | 0.280 | 0.614 | 0.381 |
| | 336 | 0.395 | 0.278 | - | - | 0.425 | 0.284 | 0.416 | 0.290 | 0.396 | 0.277 | 0.424 | 0.282 | 0.407 | 0.272 | 0.396 | 0.275 | 0.505 | 0.276 | 0.626 | 0.332 | 0.417 | 0.286 | 0.627 | 0.389 |
| | 720 | 0.431 | 0.288 | - | - | 0.455 | 0.302 | 0.454 | 0.312 | 0.445 | 0.308 | 0.452 | 0.298 | 0.462 | 0.316 | 0.435 | 0.295 | 0.552 | 0.301 | 0.635 | 0.340 | 0.454 | 0.308 | 0.646 | 0.394 |
| $1^{st}$ Count | | 22 | 23 | 5 | 6 | 3 | 0 | 0 | 0 | 0 | 0 | 1 | 2 | 1 | 2 | 0 | 0 | 1 | 1 | 0 | 0 | 0 | 0 | 0 | 0 |

Table 11: Full results for the classification task. ∗. in the Transformers indicates the name of ∗former. We report the classification accuracy (%) as the result. Higher accuracies indicate better performance. **Red**: the best, Blue: the 2nd best.

| Datasets / Models | Classical methods | | | RNN | | | Transformers | | | | | | | | | | MLP | | | CNN | | |
|---|---|---|---|---|---|---|---|---|---|---|---|---|---|---|---|---|---|---|---|---|---|---|
| | DTW (1994) | XGBoost (2016) | Rocket (2020) | LSTM (1997) | LSTNet (2018) | LSSL (2022) | Trans. (2017) | Re. (2020) | In. (2021) | Pyra. (2022) | Auto. (2021) | Station. (2022) | FED. (2022) | ETS. (2022) | Flow. (2022) | iTrans. (2024) | DLinear (2023) | LightTS. (2022) | TiDE (2022) | TCN (2019) | TimesNet (2023) | PatchMoE (ours) |
| EthanolConcentration | 32.3 | 43.7 | 45.2 | 32.3 | 39.9 | 31.1 | 32.7 | 31.9 | 31.6 | 30.8 | 31.6 | 32.7 | 28.1 | 31.2 | 33.8 | 28.1 | 32.6 | 29.7 | 27.1 | 28.9 | 35.7 | 32.8 |
| FaceDetection | 52.9 | 63.3 | 64.7 | 57.7 | 65.7 | 66.7 | 67.3 | 68.6 | 67.0 | 65.7 | 68.4 | 68.0 | 66.0 | 66.3 | 67.6 | 66.3 | 68.0 | 67.5 | 65.3 | 52.8 | 68.6 | 69.3 |
| Handwriting | 28.6 | 15.8 | 58.8 | 15.2 | 25.8 | 24.6 | 32.0 | 27.4 | 32.8 | 29.4 | 36.7 | 31.6 | 28.0 | 32.5 | 33.8 | 24.2 | 27.0 | 26.1 | 23.2 | 53.3 | 32.1 | 30.4 |
| Heartbeat | 71.7 | 73.2 | 72.2 | 77.1 | 72.7 | 75.6 | 76.1 | 77.1 | 80.5 | 75.6 | 74.6 | 73.7 | 73.7 | 71.2 | 77.6 | 75.6 | 75.1 | 75.1 | 74.6 | 75.6 | 78.0 | 77.2 |
| JapaneseVowels | 94.9 | 86.5 | 96.2 | 79.7 | 98.1 | 98.4 | 98.7 | 97.8 | 98.9 | 98.4 | 96.2 | 99.2 | 98.4 | 95.9 | 98.9 | 96.6 | 96.2 | 96.2 | 95.6 | 98.9 | 98.4 | 97.0 |
| PEMS-SF | 71.1 | 98.3 | 75.1 | 39.9 | 86.7 | 86.1 | 82.1 | 82.7 | 81.5 | 83.2 | 82.7 | 87.3 | 80.9 | 86.0 | 83.8 | 87.9 | 75.1 | 88.4 | 86.9 | 68.8 | 89.6 | 88.4 |
| SelfRegulationSCP1 | 77.7 | 84.6 | 90.8 | 68.9 | 84.0 | 90.8 | 92.2 | 90.4 | 90.1 | 88.1 | 84.0 | 89.4 | 88.7 | 89.6 | 92.5 | 90.2 | 87.3 | 89.8 | 89.2 | 84.6 | 91.8 | 92.6 |
| SelfRegulationSCP2 | 53.9 | 48.9 | 53.3 | 46.6 | 52.8 | 52.2 | 53.9 | 56.7 | 53.3 | 53.3 | 50.6 | 57.2 | 54.4 | 55.0 | 56.1 | 54.4 | 50.5 | 51.1 | 53.4 | 55.6 | 57.2 | 65.6 |
| SpokenArabicDigits | 96.3 | 69.6 | 71.2 | 31.9 | 100 | 100 | 98.4 | 97.0 | 100 | 99.6 | 100 | 100 | 100 | 98.8 | 96.0 | 81.4 | 100 | 100 | 98.9 | 95.6 | 99.0 | 99.8 |
| UWaveGestureLibrary | 90.3 | 75.9 | 94.4 | 41.2 | 87.8 | 85.9 | 85.6 | 85.6 | 85.6 | 83.4 | 85.9 | 87.5 | 85.3 | 85.0 | 86.6 | 85.9 | 82.1 | 80.3 | 84.9 | 88.4 | 85.3 | 88.8 |
| Average Accuracy | 67.0 | 66.0 | 72.5 | 48.6 | 71.8 | 70.9 | 70.3 | 71.9 | 71.5 | 72.1 | 70.8 | 71.1 | 72.7 | 70.7 | 71.0 | 73.0 | 70.5 | 67.5 | 70.4 | 69.5 | 73.6 | 74.11 |

Table 12: Anomaly detection results. Higher Affiliated-F1 (F) and AUC-ROC (AUC) values indicate better performance. **Red**: the best, Blue: the 2nd best.

| Datasets | CalIt2 | | Credit | | GECCO | | Genesis | | MSL | | NYC | | PSM | | SMAP | | SMD | | 1st Count | |
|---|---|---|---|---|---|---|---|---|---|---|---|---|---|---|---|---|---|---|---|---|
| Metrics | F | AUC | F | AUC | F | AUC | F | AUC | F | AUC | F | AUC | F | AUC | F | AUC | F | AUC | F | AUC |
| Ocsvm [1999] | 0.783 | 0.804 | 0.714 | 0.953 | 0.666 | 0.804 | 0.677 | 0.733 | 0.641 | 0.524 | 0.667 | 0.456 | 0.531 | 0.619 | 0.503 | 0.487 | 0.742 | 0.679 | 0 | 0 |
| PCA [2003] | 0.768 | 0.790 | 0.710 | 0.871 | 0.785 | 0.711 | 0.814 | 0.815 | 0.678 | 0.552 | 0.680 | 0.666 | 0.702 | 0.648 | 0.505 | 0.396 | 0.738 | 0.679 | 0 | 0 |
| Isolation Forest [2008] | 0.402 | 0.775 | 0.634 | 0.860 | 0.424 | 0.619 | 0.788 | 0.549 | 0.584 | 0.524 | 0.648 | 0.475 | 0.620 | 0.542 | 0.512 | 0.487 | 0.626 | 0.664 | 0 | 0 |
| HBOS [2012] | 0.756 | 0.798 | 0.695 | 0.951 | 0.708 | 0.557 | 0.498 | 0.124 | 0.680 | 0.574 | 0.675 | 0.446 | 0.658 | 0.620 | 0.509 | **0.585** | 0.629 | 0.626 | 0 | 1 |
| Autoencoder [2014] | 0.587 | 0.767 | 0.561 | 0.909 | 0.823 | 0.769 | 0.854 | 0.931 | 0.625 | 0.562 | 0.689 | 0.504 | 0.707 | 0.650 | 0.463 | 0.522 | 0.120 | 0.774 | 0 | 0 |
| ATransformer [2022] | 0.688 | 0.491 | 0.646 | 0.533 | 0.782 | 0.516 | 0.715 | 0.472 | 0.685 | 0.508 | 0.691 | 0.499 | 0.654 | 0.498 | **0.703** | 0.504 | 0.704 | 0.309 | 1 | 0 |
| FEDformer [2022] | 0.788 | 0.707 | 0.683 | 0.825 | 0.900 | 0.709 | 0.893 | 0.802 | 0.726 | 0.561 | 0.691 | 0.725 | 0.761 | **0.679** | 0.658 | 0.474 | 0.782 | 0.650 | 0 | 1 |
| DCdetector [2023] | 0.673 | 0.527 | 0.610 | 0.504 | 0.671 | 0.555 | 0.776 | 0.507 | 0.683 | 0.504 | 0.698 | 0.528 | 0.662 | 0.499 | 0.701 | 0.516 | 0.675 | 0.500 | 0 | 0 |
| NLinear [2023] | 0.757 | 0.695 | 0.742 | 0.948 | 0.882 | 0.936 | 0.829 | 0.755 | 0.723 | 0.592 | 0.819 | 0.671 | 0.843 | 0.585 | 0.601 | 0.434 | 0.844 | 0.738 | 0 | 0 |
| DLinear [2023] | 0.793 | 0.752 | 0.738 | 0.954 | 0.893 | 0.947 | 0.856 | 0.696 | 0.725 | 0.624 | 0.828 | 0.768 | 0.831 | 0.580 | 0.616 | 0.397 | 0.841 | 0.728 | 0 | 0 |
| TimesNet [2023] | 0.794 | 0.771 | 0.744 | 0.958 | 0.897 | 0.964 | 0.864 | 0.913 | 0.734 | 0.613 | 0.794 | 0.791 | 0.842 | 0.592 | 0.638 | 0.453 | 0.833 | 0.766 | 0 | 0 |
| Crossformer [2023] | 0.789 | 0.798 | 0.720 | 0.951 | 0.897 | 0.770 | 0.865 | 0.755 | 0.733 | 0.587 | 0.692 | 0.679 | 0.789 | 0.654 | 0.627 | 0.383 | 0.839 | 0.710 | 0 | 0 |
| PatchTST [2023] | 0.660 | 0.808 | 0.746 | 0.957 | 0.906 | 0.949 | 0.856 | 0.685 | 0.723 | 0.637 | 0.776 | 0.709 | 0.831 | 0.586 | 0.606 | 0.448 | 0.845 | 0.736 | 0 | 0 |
| ModernTCN [2024] | 0.780 | 0.676 | 0.744 | 0.957 | 0.899 | 0.954 | 0.833 | 0.676 | 0.726 | 0.633 | 0.769 | 0.466 | 0.825 | 0.592 | 0.635 | 0.455 | 0.840 | 0.722 | 0 | 0 |
| iTransformer [2024] | 0.812 | 0.791 | 0.713 | 0.934 | 0.839 | 0.794 | 0.891 | 0.690 | 0.710 | 0.611 | 0.684 | 0.640 | 0.853 | 0.592 | 0.587 | 0.409 | 0.827 | 0.745 | 0 | 0 |
| DualTF [2025] | 0.751 | 0.643 | 0.663 | 0.703 | 0.701 | 0.714 | 0.810 | **0.937** | 0.588 | 0.585 | 0.708 | 0.633 | 0.725 | 0.600 | 0.674 | 0.478 | 0.679 | 0.631 | 0 | 0 |
| CATCH [2025] | 0.835 | 0.838 | 0.750 | 0.958 | 0.908 | 0.970 | 0.896 | **0.974** | 0.740 | **0.664** | **0.994** | 0.816 | **0.859** | 0.652 | 0.699 | 0.504 | 0.847 | 0.811 | 2 | 2 |
| PatchMoE [ours] | **0.842** | **0.861** | **0.754** | **0.959** | **0.914** | **0.979** | **0.903** | 0.862 | **0.746** | 0.641 | 0.973 | **0.833** | 0.850 | 0.645 | 0.669 | 0.489 | **0.868** | **0.831** | 6 | 5 |

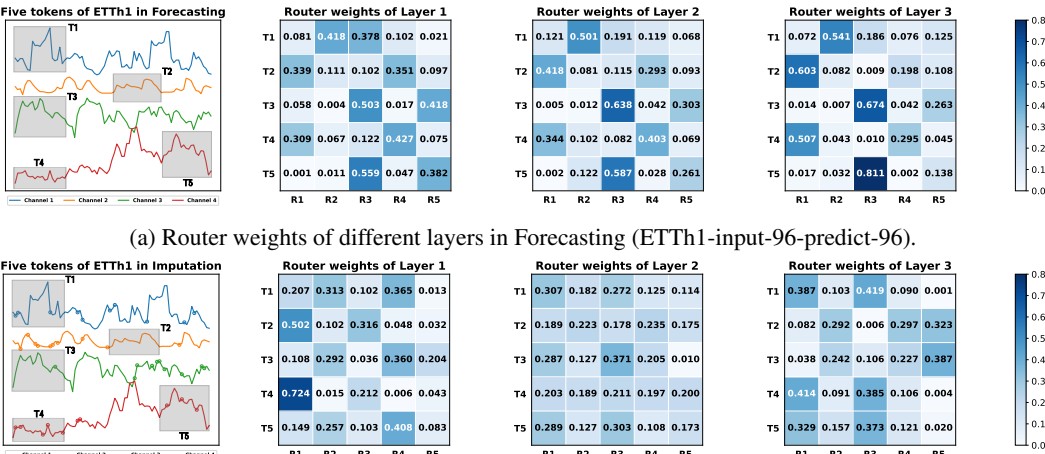

(a) Router weights of different layers in Forecasting (ETTh1-input-96-predict-96).

(b) Router weights of different layers in Imputation (ETTh1-mask-ratio-12.5%). Masked points are circled.

Figure 6: Router weights of different layers in ETTh1 (input-96), under tasks of Forecasting (horizon-96), and Imputation (mask-ratio-12.5%). We select five tokens (T1–T5) from four channels as examples to demonstrate the effectiveness of RNG-Router (with $N_r = 5$ routed experts (R1–R5)). In Forecasting, the routing strategies keep consistent from Layer 1–3, forming three clusters to capture the temporal and channel correlations, i.e., T1 itself, {T2, T4}, and {T3, T5}, which mainly relies on the shallow representations. In imputation, the routing strategies vary across layers, tuning the shallow clusters, i.e., {T1, T3, T5}, and {T2, T4}, to deep clusters, i.e., {T1, T4, T5}, and {T2, T3}, which relies more on deep representations.

## C  MODEL ANALYSIS

### C.1  REPRESENTATION ANALYSIS

As the core component in PatchMoE, the RNG-Router is designed for task-specific purposes. To further evaluate its impact, we make a special representation analysis on this routing mechanism–see Figure 6. We select five tokens from ETTh1 and track their routing weights across different MoE layers under tasks of forecasting and imputation. Since advanced task-specific models tend to implicitly utilize the shallow representations in forecasting (reflected in high CKA similarities), and deep representations in imputation (reflected in low CKA similarities), our proposed PatchMoE provides explict evidences of this capability. In Figure 6 (a), token T3 and T5, T2 and T4 are similar, and T1 is a bit similar to T3. The routing weights across three MoE layers reflect that the RNG-Router gradually achieves the clustering of tokens with similar shallow patterns, where tokens in the same cluster share the same experts. On the other hand, the imputation task relies more on high-level semantics in deep representations. It is observed that the RNG-Router gradually tunes the routing weights in deeper layers and mines the appropriate high-level correlations among representations. These evidences demonstrate that RNG-Router can effectively utilize the hierarchical representations to boost the routing of time series tokens for distinct downstream tasks, which leads to an elegant and general representation learning framework with task-specific capabilities.

### C.2  FULL PARAMETER SENSITIVITY

We conduct more analytics of PatchMoE in this section. We study the parameter sensitivity of PatchMoE–see Figure 7. Figure 7a shows that PatchMoE keep stable performance under different patch sizs, and we often choose 16 and 24 as common configurations. As the Look Back Window extends–see Figure 7b, the forecasting performance keeps consistent improvement, showing scability. Figure 7c and Figure 7d show the influences of MoE layers and routed experts, which determine model's capability of modeling the task-specific temporal and channel correlations. Results show that more MoE layers and routed experts leads to larger model capacity on large datasets like Solar and Traffic, but may cause over-fittling dilemma in small datasets like ETTh1 and ETTm2. To make accruacy and efficiency meet, we choose $L = 3$ and $N^r = 10$ as the common setting, and set 3 as the Top-K number. We also set $N^s = 1$ shared expert to extract the common patterns.

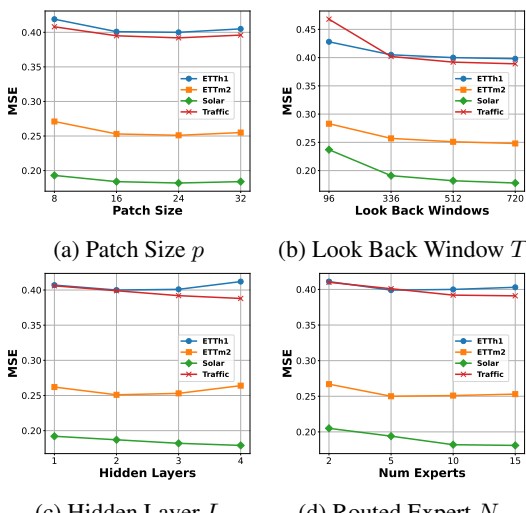

(a) Patch Size $p$  (b) Look Back Window $T$

(c) Hidden Layer $L$  (d) Routed Expert $N_r$

Figure 7: Parameter sensitivity studies of main hyper-parameters in PatchMoE, including Patch Size $p$, Length of Look Back Window $T$, number of Hidden Layers $L$, and number of Routed Experts $N_r$.

## D  FULL ABLATIONS

We list the full results of ablation studies in Table 13. It is observed that each component is very important. Without the RNG-Router, the traditional router cannot utilize the task-specific information across hierarchical representations, causing performance crash. Without Shared Experts, the model lacks capacity and performs poorly at large datasets like Solar and Traffic. Without Temporal & Channel Load Balancing Loss, the model also cannot well model the intricate temporal and channel correlations.

Table 13: Full ablation studies on key components of PatchMoE, including RNG-Router, Shared Experts, and Temporal & Channel Load Balancing Loss.

| Models | | w/o RNG-Router | | w/o Shared Experts | | w/o Loss | | PatchMoE | |
|---|---|---|---|---|---|---|---|---|---|
| Metrics | | MSE | MAE | MSE | MAE | MSE | MAE | MSE | MAE |
| ETTh1 | 96 | 0.368 | 0.394 | 0.358 | 0.393 | 0.357 | 0.392 | **0.355** | **0.390** |
| | 192 | 0.425 | 0.441 | 0.402 | 0.418 | 0.404 | 0.420 | **0.398** | **0.417** |
| | 336 | 0.432 | 0.439 | 0.437 | 0.446 | 0.420 | 0.433 | **0.418** | **0.431** |
| | 720 | 0.443 | 0.462 | 0.450 | 0.477 | 0.431 | 0.460 | **0.430** | **0.456** |
| | avg | 0.417 | 0.434 | 0.412 | 0.434 | 0.403 | 0.426 | **0.400** | **0.424** |
| ETTm2 | 96 | 0.171 | 0.262 | 0.167 | 0.254 | 0.163 | 0.247 | **0.160** | **0.244** |
| | 192 | 0.217 | 0.286 | 0.220 | 0.287 | 0.223 | 0.291 | **0.217** | **0.285** |
| | 336 | 0.289 | 0.336 | 0.283 | 0.331 | 0.275 | 0.326 | **0.273** | **0.322** |
| | 720 | 0.362 | 0.379 | 0.359 | 0.378 | 0.365 | 0.379 | **0.355** | **0.373** |
| | avg | 0.260 | 0.316 | 0.257 | 0.313 | 0.257 | 0.311 | **0.251** | **0.306** |
| Solar | 96 | 0.175 | 0.217 | 0.169 | 0.211 | 0.168 | 0.209 | **0.166** | **0.207** |
| | 192 | 0.198 | 0.223 | 0.183 | 0.228 | 0.183 | 0.228 | **0.178** | **0.222** |
| | 336 | 0.205 | 0.229 | 0.197 | 0.232 | 0.188 | 0.227 | **0.184** | **0.224** |
| | 720 | 0.210 | 0.244 | 0.202 | 0.240 | 0.200 | 0.240 | **0.198** | **0.234** |
| | avg | 0.197 | 0.228 | 0.188 | 0.228 | 0.185 | 0.226 | **0.182** | **0.222** |
| Traffic | 96 | 0.373 | 0.266 | 0.368 | 0.265 | 0.376 | 0.272 | **0.361** | **0.261** |
| | 192 | 0.386 | 0.269 | 0.420 | 0.294 | 0.392 | 0.279 | **0.382** | **0.268** |
| | 336 | 0.396 | 0.275 | 0.432 | 0.298 | 0.405 | 0.288 | **0.395** | **0.278** |
| | 720 | 0.435 | 0.295 | 0.465 | 0.313 | 0.437 | 0.299 | **0.431** | **0.288** |
| | avg | 0.398 | 0.276 | 0.421 | 0.293 | 0.403 | 0.285 | **0.392** | **0.274** |

