# OpenReview forum: "Unlocking the Power of Mixture-of-Experts for Task-Aware Time Series Analytics"
_ICLR.cc/2026/Conference — ICLR 2026 Conference Withdrawn Submission_

### Official Review · Reviewer_Zbts · 2025-10-28

**Soundness:** 3
**Presentation:** 2
**Contribution:** 3
**Rating:** 4
**Confidence:** 4

**Summary:**

This paper introduces PatchMoE, a task-aware Mixture-of-Experts framework tailored for time series analysis. Unlike traditional MoE architectures, PatchMoE incorporates a Recurrent Noisy Gating mechanism to leverage hierarchical, task-specific representations across different layers. The routing process is guided by a Temporal & Channel Load Balancing Loss to model sparse and intricate correlations. The framework is designed to support multiple downstream tasks such as forecasting, anomaly detection, imputation, and classification. Experimental results on various datasets demonstrate PatchMoE’s superior ability to model task-specific temporal and channel relationships, showcasing its effectiveness.

**Strengths:**

1. This paper conducts extensive experiments on various tasks and datasets to verify the effectiveness of the proposed model.
2. The paper is well-structured and easy to follow.

**Weaknesses:**

1. Although the starting point of this paper is to achieve task-aware modeling, the proposed method is rather implicit: the CKA similarity varies across different tasks, so the model needs to consider multi-layer information during routing. I think that considering multi-layer information is not directly related to task awareness, and the paper does not provide any experiments showing whether the CKA similarity changes accordingly when using the proposed method.
2. The proposed method contains many components, while the ablation study is too coarse to validate their effectiveness:
 - How is RNG-router compared with the original router in NLP using a simple linear layer?
 - Why use sampling in routing instead of directly taking topK?
3. Some technical details are not so clear:
 1. In Equation (3), is MSA applied in a channel-independent way?
 2. In Equation (4), how is GRU cell applied? $(N \times n)$ is merged with batch dimension or with latent dimension?
 3. Despite being applied to temporal and channel separately, the original source of the balancing loss should be explicitly cited.

**Questions:**

1. In my understanding, the proposed model is still channel-independent: 1. MSA is applied to each channel independently; 2.the RNG-based MOE only takes one token (and the same token in previous layers) as input. How does this capture cross-channel dependency?
2. As the original motivation of MOE is to replace dense FFN layers for efficiency. How is the efficiency of patchMOE compared with dense layer and original MOE?

---

### Official Review · Reviewer_kR3k · 2025-10-28

**Soundness:** 4
**Presentation:** 3
**Contribution:** 3
**Rating:** 6
**Confidence:** 4

**Summary:**

This paper presents PatchMoE, a novel Mixture-of-Experts (MoE) framework tailored for time series analysis. The key contributions are as follows:

1.	Recurrent Noisy Gating (RNG - Router): Employs hierarchical task-specific routing to explore cross-layer representational variances;
2.	Temporal & Channel Load Balancing Loss: Promotes sparse and balanced utilization of experts to capture intricate temporal and channel correlations;
3.	Hybrid Design of Shared and Routed Experts: Facilitates the modeling of both general patterns and task-specific dynamics.

Based on the mentioned characteristics, PatchMoE is a flexible and general architecture that effectively enhances the capabilities of the temporal Transformers.

**Strengths:**

**S1**
  A recurrent noisy router, which is shared across layers and conditions routing on the representations of previous layers through a GRU, represents a judicious approach to integrating hierarchical information into the routing process. The paper explicitly states that during training, routing scores are sampled in the continuous score space via [specific method], and at inference, these scores are replaced by [specific method]. This strategy is well-suited for stabilizing top-gating.

**S2**
  The Temporal & Channel Load Balancing Loss is designed to prevent expert collapse and promote structured sparsity along the time and channel axes. This is a valuable objective when dealing with multivariate data, especially when maintaining a channel - independent backbone.

**S3**
  The combination of shared experts, which are responsible for general patterns, and routed experts, which are for specialization, aligns with the best practices in sparse Mixture - of - Experts (MoE) systems. This approach also aids in managing the trade-off between capacity and specialization.

**Weaknesses:**

**W1**
   The paper appears to adopt non-overlapping patches by default. Nevertheless, previous research indicates that overlapping patches with stride can efficiently mitigate information loss and yield superior performance.
   - The authors are expected to elucidate the reasons for choosing non-overlapping patches, such as whether it is due to efficiency, model stability, or other factors.
   - Incorporating a comparative experiment (comparing overlapping and non-overlapping patches) would further enhance the work, thereby justifying this design decision.

**W2**
 - Provide the full training objective $ L _{\text{total}} = L _{\text{task}} + \lambda L _{\text{bal}} $, the values or schedules for $ \lambda $, $ \alpha $, $ \beta $ per task/dataset, and a brief sensitivity study. This will clarify how strongly the balancing terms influence optimization across tasks.

**W3**
   - Hidden state initialization and management: the router is shared across layers, but the initialization of $h _0$ per input, whether it is reset per sequence or carried across batches, and any truncation policy are not described.
   - Task heads and losses for anomaly detection, imputation, and classification are not detailed in the main text, making it hard to attribute gains to the backbone versus the heads.

**W4**
   Typos and table annotations (e.g., “Bond” should be “Bold”; “Routher” should be “Router”; “ETThm2” vs. “ETTm2”; “N^r” vs. ) and a few inconsistent variable names hamper readability.

**Questions:**

See W1-W4.

---

### Official Review · Reviewer_Pee9 · 2025-10-31

**Soundness:** 3
**Presentation:** 2
**Contribution:** 2
**Rating:** 4
**Confidence:** 4

**Summary:**

This paper proposes PatchMoE, a Mixture-of-Experts (MoE) framework designed for task-aware time series analytics. Unlike traditional MoE models that are task-agnostic, PatchMoE introduces Recurrent Noisy Gating (RNG-Router) to incorporate hierarchical representations across layers, enabling task-specific routing decisions. It further introduces a Temporal & Channel Load Balancing Loss to capture both temporal and cross-channel correlations — a notable challenge in time-series contexts where dependencies exist in both dimensions. The model is evaluated comprehensively across five major tasks — forecasting, anomaly detection, imputation, and classification — showing consistent state-of-the-art (SOTA) performance across over 25 benchmark datasets.

**Strengths:**

1. Joint Temporal-Channel Modeling.
The Temporal & Channel Load Balancing Loss is a thoughtful addition that effectively addresses channel independence in transformers. Empirical evidence shows significant gains in multivariate datasets (e.g., Electricity, Solar), indicating the model’s ability to capture sparse cross-channel correlations without sacrificing efficiency
2. Comprehensive Evaluation.
The experiments span univariate and multivariate forecasting, anomaly detection, imputation, and classification — an unusually broad and rigorous evaluation suite. The model consistently outperforms strong baselines (iTransformer, PatchTST, TimeMixer++, etc.), demonstrating robustness across diverse domains and metrics

**Weaknesses:**

1. Limited Theoretical Analysis.
While empirical performance is strong, the paper lacks formal analysis of why RNG-Router improves routing stability or how the Temporal-Channel loss optimizes sparsity theoretically. A deeper exploration (e.g., gradient dynamics or convergence properties) would improve the conceptual depth.

2. Efficiency and Scalability Discussion Missing.
Although the authors mention model efficiency, they do not quantify computational overhead compared to PatchTST or iTransformer. Given the recurrent gating and dual balancing loss, it is unclear how well PatchMoE scales to extremely long sequences or high-channel datasets beyond those tested.

3. Limited Interpretability Discussion.
While MoE offers modular interpretability potential (expert specialization), the paper does not analyze expert behaviors or visualizations of expert routing distributions, which would strengthen the understanding of task-specific routing.

**Questions:**

1. How sensitive is PatchMoE to the number of experts (Nr) and routing sparsity (Top-k)?
2. Does RNG-Router generalize across tasks (e.g., trained jointly vs. task-specific fine-tuning)?
3. What's the impact of the different treatment of eq (7) during training and inference?

---

### Official Review · Reviewer_nh9u · 2025-11-01

**Soundness:** 3
**Presentation:** 3
**Contribution:** 3
**Rating:** 4
**Confidence:** 4

**Summary:**

This paper proposes PatchMoE, a Mixture-of-Experts (MoE) framework tailored for task-aware time series analytics. Unlike existing MoE architectures designed for text or vision, PatchMoE introduces a Recurrent Noisy Gating (RNG-Router) that dynamically leverages hierarchical layer representations to capture task-specific characteristics across various time series tasks (forecasting, anomaly detection, imputation, classification). Additionally, it incorporates a Temporal & Channel Load Balancing Loss to model sparse correlations and maintain diversity in expert routing.

**Strengths:**

- Proposes a **unified and generalizable MoE framework** for multiple time-series tasks.
- Introduces a **Recurrent Noisy Gating (RNG-Router)** to adapt routing based on hierarchical representations, improving task specificity.
- **Temporal & Channel Load Balancing Loss** effectively addresses the lack of channel correlation modeling in standard Transformers.
- Demonstrates **strong empirical results** across diverse benchmarks (forecasting, anomaly detection, imputation, classification).
- Clear ablation studies validating the contribution of each component (RNG-Router, shared experts, load-balancing loss).
- Comprehensive comparisons with modern baselines (e.g., TimeMixer, Time-MoE).

**Weaknesses:**

- [1] **Incremental novelty despite strong performance**
  - The architectural contributions, though well-motivated, largely build on known MoE principles.
  - The introduction of a recurrent router and balancing loss feels like a careful **engineering enhancement** rather than a fundamental conceptual advance .
- [2] **Performance heterogeneity across metrics and tasks**
  - While PatchMoE achieves SOTA in many tables (e.g., Table 1–3), **relative improvements vary widely** across datasets and evaluation criteria.
  - For instance, it shows **clear superiority in reconstruction-oriented metrics (MSE, MAE)** but **smaller or inconsistent gains** in discriminative metrics such as accuracy and F1.
  - **In some anomaly detection datasets (e.g., MSL, NYC)**, the margins over CATCH are marginal or reversed in AUC  .


- [3] **Overcomplex design vs. marginal efficiency analysis**
  - The proposed architecture involves **multiple MoE layers with dual experts (shared + routed)** and an RNG mechanism for every layer, potentially increasing training cost.
  - However, there is **no quantitative discussion of training or inference efficiency** compared to single-expert Transformers like iTransformer or PatchTST.

- [4] **Lack of analysis on why it works well**
  - Although the paper provides ablations (Table 13) and visualization of routing weights (Figure 6), these analyses focus only on component removal or qualitative trends.

**Questions:**

Please refer to the weaknesses above.

---

### Note · Authors · 2025-12-04

**Comment:**

I have read and agree with the venue's withdrawal policy on behalf of myself and my co-authors.

**Withdrawal Confirmation:**

I have read and agree with the venue's withdrawal policy on behalf of myself and my co-authors.